# Co-Occurrences of Forms of Child Undernutrition in India: Insights from the National Family Health Survey

**DOI:** 10.3390/nu17060977

**Published:** 2025-03-11

**Authors:** Pooja Arora, Mrigesh Bhatia, Laxmi Kant Dwivedi

**Affiliations:** 1Department of Survey Research and Data Analytics, International Institute for Population Sciences, Mumbai 400088, Maharashtra, India; kpoojak@yahoo.com (P.A.); laxmikant@iipsindia.ac.in (L.K.D.); 2Department of Health Policy, London School of Economics, London WC2A 2AE, UK

**Keywords:** co-occurrences, stunting, wasting, underweight, anaemia, child undernutrition, India, NFHS

## Abstract

*Background*: The composite index of anthropometric failure (CIAF) studies co-occurrences of three forms of child undernutrition: stunting (S), wasting (W), and underweight (U). This study attempts to modify it through the inclusion of a fourth form of undernutrition, that is, anaemia (A), serving as a proxy for micronutrient deficiencies among under-five children in India. *Methods*: Spatial and multivariate analyses were employed to analyse the co-occurrences of child undernutrition with reference to the child’s and mother’s characteristics using National Family Health Survey (NFHS) data. *Results*: The modified index of “CIAF + Anaemia” identified thirteen manifestations of child undernutrition in India, the most prevalent co-occurrence being “only anaemia” (30%), followed by a triple burden or co-occurrence of stunting, underweight, and anaemia (SUA) (12%). The prevalence of the quadruple burden of child undernutrition (SWUA) was found to be highest in the states of Jharkhand and Gujarat (7%). A higher likelihood of the co-occurrence of “SUA” was observed among underweight mothers (16%), whereas that of “only anaemia” was observed more among overweight mothers (35%) compared to their counterparts. The co-occurrences “SUA” and “SWUA” were found to be moderately clustered among the districts of India. *Conclusions*: Overall, the study reinforces the need for early identification and specialised treatment approaches for children burdened with multiple forms of undernutrition to prevent its scarring effect.

## 1. Introduction

As defined by the WHO, undernutrition has four broad sub-forms: stunting, wasting, underweight, and deficiencies in vitamins and minerals [1]. However, the phenomenon when these four forms co-occur in a child has not been studied in the literature. Svedberg (2000) came up with the composite indicator of child anthropometric failure (CIAF) to reflect the true burden of undernutrition at the individual level, studying all possible co-occurrences from combinations of stunting, wasting, and underweight [2]. Such indices help identify communities or areas where many children experience multiple failures and allocate resources to the areas of greatest need [3]. The seven different categories of CIAF, as explained by Nandy (2005), are A: “no failure”, which means children who do not suffer from any anthropometric failure; B: “wasting only”, which means children with an acceptable weight and height for their age but who have subnormal weight for their height; C: “wasting and underweight”, which means children with above-normal heights but whose weight for their age and weight for their height are too low; D: “wasting, stunting, and underweight”, which means children who suffer from anthropometric failure on all three measures; E: “stunting and underweight”, which means children with a low weight for their age and a low height for their age but who have an acceptable weight for their height; F: “stunting only”, which means children with a low height for age but who have an acceptable weight, both for their age and for their short height; and, lastly, Y: “underweight only”, which means children who are only underweight [4]. Categories B: “wasting only”, F: “stunting only”, and Y: “underweight only”, are also termed as standalone conditions.

Quoted from a 2019 report by the Government of India, Figure 1 is a Venn diagram showing the co-existence of multiple forms of undernutrition among under-five children in India using NFHS 2015–2016 data [5]. Among children aged 0–5 years, 6.5% of children were stunted and wasted, as well as underweight; 18.4% of children were stunted and underweight, and 8.2% of children were wasted and underweight. It revealed that after disaggregating the co-existence of these three conditions, 13.6% of children were only stunted (against a 38.4% overall prevalence of stunting), 2.6% were only underweight (against a 35.7% overall prevalence of underweight), and 6.3% (against a 21% overall prevalence of wasting) were wasted. State-wise analysis showed that Jharkhand (10.9%) led with the highest prevalence of multiple burdens of malnutrition encompassing stunting, wasting, and underweight conditions among under-five children, followed by Madhya Pradesh (8.5%) and Bihar (8.1%).

The fourth sub-form of child undernutrition, involving deficiencies in the intake of essential vitamins and minerals, commonly known as micronutrients, when grouped with stunting, wasting, and underweight, can help understand the real burden of undernutrition. In many cases where diets are poor in micronutrients, multiple micronutrient deficiencies are likely to affect the development of anaemia synergistically [6]. The WHO defines anaemia in children aged under 5 years as a haemoglobin concentration <110 g/L at sea level and is associated with increased risks for child mortality, negatively impacting the cognitive and physical development of children. If we take anaemia as a proxy for representing a lack of essential vitamins and minerals, there are many instances in the literature that analyse different forms of child undernutrition interacting or co-occurring with one another [7]. For example, a child cannot be stunted and wasted at the same time without underweight, as it is physically not possible since there is a high effect of underweight on the concurrence of both [8]. A recent study showed that stunting and wasting are linked and that wasting can increase the risk of subsequent stunting [9]. Another study tried to assess the role of underweight in predicting the stunting status of the child using CNNS 2016–2018 data and concluded that it could be used as a substitute for surveys that poorly measured length or height data [10].

Similarly, essential micronutrients play a vital role in fostering physical growth as they help produce various necessary enzymes and hormones that aid in regulating biological processes in our body [11]. Mohammed (2019) found a high level of CAS (concurrent anaemia and stunting) among infants and young children in Ethiopia that was associated with various dietary and non-dietary factors [12]. Some studies have also examined the concurrences of anaemia and different growth indicators at the population level. A recently published article analysed clustering in Indian districts of 19 pairs of combinations of dual burdens of different nutritional outcomes and found that several of them had significantly higher prevalence at the state and district levels [13].

Findings from the 2019 GBD study by the ICMR suggest that the malnutrition targets set by India’s National Nutrition Mission (NNM) are aspirational, and the rate of improvement needed to achieve these targets is much higher than the rate observed presently, which might be challenging to reach in a short period [14]. Child growth anthropometric failures and anaemia are the most commonly used indicators for assessing the nutritional status of children. Still, their overlapping is rarely studied. A better understanding of clustering between different nutritional outcomes is required, as co-existing forms of malnutrition (CFM) in any form result in a heightened risk of distinct health adversities, diverging from comparable standalone conditions [15]. Additionally, a shift in how child undernutrition is understood and managed is urgently needed if the World Health Assembly and the UN Sustainable Development Goal targets are to be met.

Similar to the dual or multiple burdens of malnutrition when two or more forms of malnutrition co-occur within individuals, this study tried to identify the single, dual, triple, and quadruple burdens of child undernutrition by studying the co-occurrences of stunting, wasting, underweight, and anaemia among under-five children in India. Spatial analysis of such co-occurrences at the district level can help the government plan interventions as per the intensity of different co-occurrences across geographies. Therefore, this study tried to understand the clustering of child undernutrition outcomes at the individual as well as the population levels.

## 2. Materials and Methods

### 2.1. Data Source and Study Population

Data from the National Family Health Survey (NFHS), which is the Indian version of the Demographic Health Survey (DHS), was used in this study. The NFHS is a popularly known household survey in India that collects demographic and health-related information every few years from respondents, including women, men, and children and household members with specialised tools for each of these types of respondents. It also collects blood samples and provides information on biomarkers like height, weight, haemoglobin, etc. Up until now, data from five survey rounds have been collected, and the latest round, NFHS-5 (2019–2021), has collected information from over 636,699 households. The NFHS uses a multistage stratified sampling approach for producing district and state-level estimates, and their detailed sampling strategy can be accessed from their published reports.

For the present study, the Round V 2019–2021 data of NFHS were mainly analysed. Round III 2005–2006 and Round IV 2015–2016 datasets of NFHS were used for comparison purposes to observe trends. Data from the Comprehensive National Nutrition Survey (CNNS) were also used in a few instances, but only for descriptive statistics since the estimates are not representative at the district level. Regarding software, STATA version 15 was used for analysing all the data. Since this study is based on secondary data from various surveys conducted in India, these datasets have been archived in a public repository; therefore, the data are easily accessible, and ethical approval was not needed to conduct this study. Sampling weights were used for all computations and analysis.

The NFHS, through its women’s, biomarker’s, and household’s questionnaires, collects information on multiple domains like birth history, household amenities, source of toilet facilities, type of cooking fuel used in the household, education, etc. Since our outcome variables involved the use of data on height, weight, age, haemoglobin, and WHO-suggested z-scores like height-for-age z-scores (HAZ), weight-for-height z-scores (WHZ), and weight-for-age z-scores (WAZ), the plausible cases which were flagged as “out of the permitted ranges” in the NFHS dataset were dropped. In total, NFHS-5 (2019–2021) collected information from around 198,475 children aged 6–59 months, as accessed from the “kids” file of the NFHS from the publicly available datasets on the website dhsprogram.com on 19 March 2023. After excluding all the children whose information was missing or plausible for the outcome variable, our study performed an analysis on the final sample size of 175,289 children aged 6–59 months. Children aged 0–5 months were not included since information on anaemia was not collected for them.

### 2.2. Computing the Outcome Variable, Modified CIAF + Anaemia

As described earlier, the traditional CIAF utilises information on the stunting, wasting, and underweight status of each child and categorises children into seven categories in such a way that we get to know who are the children who are free from any form of undernutrition (i.e., none), who are the ones suffering from only one form or standalone forms (that is, only stunting, only wasting, or only underweight), who are the ones suffering from any two of the three forms of undernutrition (e.g., wasting and underweight, stunting and underweight), and who are the ones who are suffering from all three forms of undernutrition (stunting, wasting, and underweight). These co-occurrences are also termed as anthropometric failures (AFs). A modified version of the CIAF was computed in the same way while taking into consideration the status of anaemia as well. The Table 1 describes the fourteen mutually exclusive categories of our outcome variable, or the four major types of burdens or co-occurrences of child undernutrition in which children under the age of five could be categorised.

### 2.3. Analysis

As part of the descriptive statistics, the trend of all four undernutrition outcomes, in general, was studied across NFHS survey rounds and the CNNS. The co-occurrences at the individual level were observed, and, as listed in Table 1, the prevalence of the fourteen manifestations of child undernutrition was computed. Out of the fourteen, the trend of the most prevalent co-occurrences was studied across NFHS survey rounds. Particularly, the pattern of quadruple burden of child undernutrition, that is, when the child is suffering from all four forms of undernutrition, was also studied with respect to state and the child’s age in months using the latest round data of NFHS. District-level prevalence of the same was represented through a spatial map.

With the help of the literature and data available, different explanatory variables were selected, and their distribution was analysed as part of the univariate analysis or sample description using data from NFHS-5 (2019–2021). For the bivariate analysis, all the types of burdens or co-occurrences of child undernutrition were studied across various background characteristics and maternal and child characteristics. For assessing the association between the outcome variable and the covariates, the chi-square test of association was used for testing statistical significance. Ordered logistic regression was run for the outcome variable with 5 categories (no burden, single burden, dual burden, triple burden, and quadruple burden for child undernutrition) to identify determinants of burdens of child undernutrition in India. Further, binary logistic regression was also run to confirm the determinants of some of the most prevalent co-occurrences while coding the co-occurrences into binary format, 1 representing “yes” for the co-occurrence and 0 representing “no” for the remaining children. A variable description for each independent variable considered is given in Table 2.

For analysing clustering at the population level, spatial analysis across districts was conducted for the data of NFHS Round V 2019–2021 using the GeoDa software v1.18. Shapefiles were accessed from dhsprogram.com, and spatial autocorrelation using univariate local Moran’s I statistic is computed for some of the most prevalent burdens to study how clustered they are at the district level. The Moran’s I statistic lies between the range of −1 and 1 and could be defined as the effect of an attribute of one district on that of its neighbouring districts. It is measured as the slope of the regression run on the spatial lag of the variable with respect to the variable itself, where the spatial lag of a specified variable is computed by taking the weighted average of the variable of the neighbouring districts. Values of Moran’s I greater than 0 suggested clustering of similar values, that is, high values near high values or low values near low values. Values less than 0 indicate spatial dispersion, that is, high values surrounded by low values and vice versa. Values near 0 suggest a random spatial pattern, meaning no significant clustering. A pseudo *p*-value is computed for testing the significance of each value of Moran’s I statistic using the permutation-based randomisation function of GeoDa. Queen contiguity weights were used, which define the neighbours of a district as all the adjacent districts that share a common edge or common vertex with it.

For comparison between the magnitude of the district-level clustering of the existing indicators of the four forms of child undernutrition outcomes, the co-occurrences from the traditional CIAF and those from the modified CIAF + anaemia version, a univariate local Moran’s I statistic was also computed for them. For example, a comparison in district-level clustering of the nutrition failure or anthropometric failure of “only stunting” or “none” is possible, since both these categories are observed in the traditional CIAF as well as the modified CIAF + anaemia. Spatial autocorrelation was not computed for other categories or co-occurrences as their district-level prevalences were quite low.

Moving forward in the comparison, bivariate Moran’s I is also analysed to study the co-clustering between existing indicators of forms of child undernutrition outcomes, the co-occurrences from the traditional CIAF, and those from the modified CIAF + anaemia. Bivariate Moran’s I statistics show the degree of spatial association between the spatial lag of one variable with respect to another variable. Again, only some of the co-occurrences are studied that were found to be more prevalent. Similar to univariate local Moran’s I, the bivariate local Moran’s I measures whether high (or low) values of one variable in a given location are associated with high (or low) values of another variable in neighbouring locations.

Taking the significant explanatory variables found from the analysis conducted for studying co-occurrences at the individual level, spatial analysis is particularly conducted for identifying the causes behind the clusters of the quadruple burden of child undernutrition. District-level prevalences are computed for the explanatory variables, and the quadruple burden of child undernutrition and univariate and bivariate Moran’s I are computed. Bivariate LISA (local indicator of spatial association) maps are also generated to assess the hotspots of the quadruple burden of child undernutrition alongside the explanatory variables. Hotspots refer to the areas where high/high clustering is observed; cold spots are where low/low clustering is observed, and spatial outliers or poor clustering is observed where high/low and low/high values occur alongside the neighbouring districts.

Taking selected explanatory variables from the findings of bivariate local Moran’s I, spatial regression was conducted to assess the spatial determinants of the quadruple burden of child undernutrition. A spatial lag regression model was run to study how a dependent variable in a district is affected by the independent variable in that district as well as in neighbouring districts. Regression diagnostic parameters like R square and the Akaike information criterion (AIC) were computed for the selection of the spatial lag regression model. R square represents the proportion of variation explained in the outcome variable, and its value lies between zero and one, where values closer to one signify that the model fits the data well. On the other hand, lower values of the AIC signify a better fit of the model.

## 3. Results

### 3.1. Descriptive Statistics

Figure 2 shows the declining trend of different undernutrition outcomes with respect to NFHS-3, 4, 5, and the CNNS. An improvement over time was observed in stunting and underweight status, as their prevalence of 48% and 43% in NFHS-3 comes down to 36% and 32% in NFHS-5. The trend of wasting was also decreasing over time, but still, not much progress was observed, as a reduction of only 1% in wasting was evident from NFHS-3 to NFHS-5. Anaemia showed the most worrisome trend, increasing from 59% in NFHS-4 to 68% in NFHS-5. Estimates of CNNS had been nearly the same as that in NFHS-5, except for anaemia, where only 41% of children were found to be anaemic in CNNS 2016–18 compared to a 68% rate of anaemia found among children under five in NFHS-5 2019–2021.

Table 3 shows the prevalence of co-occurrences from the traditional CIAF and those from the modified CIAF + anaemia. The traditional CIAF, which shows the occurrence of stunting, wasting, and underweight in combination, when computed in NFHS-5, showed that around half of the children were not conditioned with any type of undernutrition like stunting, wasting, or underweight, whereas the other half were burdened with one, two, or three types of undernutrition deficits. Around 16% of the children were both stunted and underweight, whereas 15% of children were found to be only stunted without being wasted or underweight. These two were the most prevalent co-existent conditions, followed by a dual burden of 8% of children who were both wasted and underweight and 6% of children who were only wasted. Around 5% of the children were suffering from stunting, wasting, as well as underweight. The least prevalent condition was “underweight only” (2%).

Table 3 also shows the univariate distribution of the new index, which portrays that around 20% of children had no burden, 30% had a single burden, 20% had dual burden, 16% had triple burden, and 4% had quadruple burden of undernutrition. Figure 3 also shows the distribution of the newly formed modified index of CIAF + anaemia, analysing co-occurrences of stunting, wasting, underweight, and anaemia in a pie-chart format, emphasising the mutually exclusive nature of the categories. We find that the most prevalent co-occurrence amongst all was the standalone condition “only anaemia” (23%), followed by the co-occurrence of stunting, underweight, and anaemia (SUA), which stands at around 13% among children aged 6–59 months in India. As observed in the literature, our findings confirm that no dual burden of “stunting and wasting” exists. Similarly, no child was found to have the triple burden of the co-occurrence of “stunting”, “wasting” and “anaemia”. The three conditions co-occurred only in the presence of ”underweight”. Anaemia, being the most prevalent condition overall, was found to have the highest amount of interference with almost every other growth indicator, whereas stunting was the next dominating one, followed by underweight. The quadruple burden, meaning the presence of all undernutrition morbidities, accounted for a prevalence of 4% at the national level. There were 17% of children who were found to be free from all forms of undernutrition.

Out of the fourteen types of different co-occurrences of forms of undernutrition studied in the modified index of CIAF + anaemia, Figure 4 observes the trend of the eight most prevalent co-occurrences of undernutrition outcomes across NFHS rounds. Conditions like “only anaemia” and the co-morbidity of “stunting and anaemia” (SA) showed a decreasing trend until NFHS-4, but a spike was observed in their prevalence during NFHS-5. Consequently, the opposite was observed in the prevalence of the “none” category, that is, those free from any form of undernutrition, as it was found to be increasing from NFHS-3 to NFHS-4, but a drop was observed in NFHS-5. The trend of the co-occurrence of “stunting, underweight, and anaemia (SUA)”, “stunting, wasting, underweight, and anaemia (SWUA)”, and “stunting and underweight (SU)” was found to be decreasing. Almost no change was observed in the “only stunting” category.

### 3.2. Variation Observed in Outcome Variable with Respect to Background Characteristics

Figure 5 shows the state-wise variation in the quadruple burden of child undernutrition, that is, the co-existence of stunting, wasting, underweight status, and anaemia in children aged 6–59 months. Its prevalence ranged from 0% in Mizoram to 7% in Jharkhand. Figure 6 represents a spatial representation of the same indicator when observed district-wise. It was observed that some parts of the Northeastern, Northern, and Southern states performed well, with at least district-level prevalence of 0–1% of the quadruple burden of child undernutrition. The Eastern and Western parts of India had some of the worst-performing districts. The Dhule district had the highest prevalence of around 14% of children suffering from all four issues of undernutrition, followed by the Jalgaon district of Maharashtra (13%) and the Paschimi Singhbhum district of Jharkhand (13%).

Figure 7 observes the prevalence and trend of the four existing forms of undernutrition outcomes and the quadruple burden of child undernutrition across ages, giving us a longitudinal perspective as a child ages from 0 to 59 months using data from NFHS Round IV (2015–2016) and Round V (2019–2021). Anaemia stood out with the highest prevalence throughout the span of the first five years of a child. It stood constant at around 70% for children aged 6–23 months. The prevalence of wasted children (approximately 40%) was observed to be almost twice as much at the time of birth compared to that of stunted and underweight children (around 20%). But after the age of 1 year, children were observed to become more likely to be stunted or underweight and less likely to be wasted. Wasting showed a declining trend while stunting and underweight showed an increasing trend. This pattern was evident until the age of 2 years, after which the prevalence of all three indicators, stunting, wasting, and underweight, remained constant. Anaemia also started to decline after the age of 2 years and continued decreasing until the age of 5 years. After the age of 4 years, there were almost equal chances of around 40% for the child to be either stunted, underweight, or anaemic.

Table 4 shows the bivariate analysis, representing the prevalence of components of the newly made index of anthropometric deficits and anaemia (CIAF + anaemia) with respect to background characteristics. A chi-square test of association was performed to test the association of each of the explanatory variables with the outcome variable of the modified index “CIAF + anaemia”, and all the explanatory variables were found to be significantly associated with the outcome (*p* < 0.01). One needs to be cautious when interpreting standalone forms or single burdens, especially regarding “only wasting” and “only underweight”, as their overall prevalence at the national level was found to be relatively low, at 2% and 1%, respectively. Further, the univariate distribution of all the explanatory variables used can be accessed from Appendix A in the Appendix A to get an understanding of the sample of the study population under consideration.

In Table 4, it is observed that dual and triple burdens of undernutrition were found to be more common among children with a lower birth weight than their counterparts. In contrast, the prevalence of standalone forms like “only anaemia” and “only stunting” was found to be highest in normal-weight babies (30.5% and 4.2%, respectively). The prevalence of having multiple forms of undernutrition was highest among children with a birth order of four or more than their counterparts. At the same time, a reverse pattern was observed in the case of the standalone form “only anaemia”, as it was found to be most prevalent among children with birth order 1.

Male children were found to be more burdened with co-morbidities of undernutrition compared to female children. Lesser prevalence of the indicators “SWUA”, “WUA”, “SA”, and “only anaemia” was observed in children above two years of age, and older children were found to be more burdened with conditions like “SUA” and “SWU”. A lower prevalence of clustering of multiple occurrences of undernutrition outcomes was observed among children whose delivery took place at an institution, those who were wanted, and those with literate mothers. Co-morbidities of undernutrition were found to be higher in the case of mothers with a lower dietary intake than mothers consuming a diet richer in various food groups. The quadruple burden of undernutrition “SWUA” was found to be highest in the Western region (6.2%), followed by the Eastern and Central regions (5.1% and 3.6%, respectively). Around 29% of children were free from any form of undernutrition among those with overweight mothers, whereas the prevalence of the same was 18% among those with underweight mothers. Talking about the two most prevalent co-occurrences, the prevalence of the co-occurrence “SUA” was found to be higher among underweight mothers (15.6%), whereas that of “only anaemia” was found to be higher among overweight or obese mothers (35.1%). Children from poorer households, rural backgrounds, those using unimproved toilet facilities or exposed to open defecation, and those from the castes SC/ST/OBC were more burdened with co-occurrences of undernutrition compared to their counterparts.

Coding these co-occurrences into a binary format, separate variables were created for some of the most prevalent co-occurrences, and the findings from the bivariate analysis were confirmed in a multivariate analysis. Binary logistic regression models were run to analyse the determinants of five nutrition failures, namely, “SWUA”, “SUA”, “SA”, “only anaemia” and “only stunting”. Detailed regression results can be referred to in Appendix A of the Appendix A.

### 3.3. Studying Determinants of Burdens of Child Undernutrition in India

Table 5 presents the results of the ordered logistic regression model, analysing the outcome variables categorised into five ordered groups: none, single, dual, triple, or quadruple burdens of child undernutrition in India. The analysis is based on data from NFHS-5 (2019–2021). It was found that the dietary intake of the mother, the wealth index of the household, the sanitation facility in the household, the literacy status of the mother, the sex of the child, the age of the child, and the birth order of the child were significant predictors of the burdens of child undernutrition in India. It was found that the children belonging to the Western region of India were 68% more likely to be more burdened with multiple undernutrition failures compared to children belonging to the Northern region (AOR 1.68; *p* < 0.01). Children belonging to households that practice open defecation were 14% more likely to be more burdened with forms of undernutrition compared to households using an improved source of sanitation facility (AOR 1.14; *p* < 0.01). Children whose mothers had a higher dietary intake had lower odds of being burdened with forms of undernutrition compared to children whose mothers had a lower dietary intake (AOR 0.95; *p* < 0.05). It was discovered that children with underweight mothers had a higher likelihood of being burdened with undernutrition compared to children with normal-weight mothers (AOR 1.46; *p* < 0.01). Children who were male, born with a low birth weight, less than 2 years of age, or had a birth order of four or more had higher odds of being burdened with undernutrition compared to their counterparts. It was observed that children belonging to literate mothers, those who were delivered in a health facility, and those belonging to a higher socioeconomic status household had a lower likelihood of being burdened with multiple forms of undernutrition compared to their counterparts. An uncommon finding was also observed where children belonging to rural areas were found to be 5% less likely to be burdened with child undernutrition compared to those belonging to urban clusters (AOR 0.95; *p* < 0.05).

### 3.4. Clustering of Co-Occurrences of Forms of Child Undernutrition Across Districts of India

Table 6 represents findings from spatial analysis performed in the NFHS-5 dataset to assess spatial autocorrelation of the existing four forms of undernutrition, co-occurrences from the traditional CIAF, and co-occurrences from the modified CIAF + anaemia version across districts in India. Amongst the four child undernutrition outcomes of stunting, wasting, underweight, and anaemia, the most substantial clustering across districts was found in the case of underweight (Moran’s I = 0.677; *p* < 0.01), followed by anaemia, stunting, and wasting (Moran’s I = 0.579, 0.511, and 0.463, respectively). Only selected types of co-occurrences were kept in the table for meaningful comparison across the co-occurrences or nutrition failures from the two indices of CIAF and CIAF + anaemia. The co-occurrences involving the presence of stunting and anaemia were selected as they were the most dominating and relevant ones. The comparison in the magnitude of clustering of co-occurrences from both indices enables us to infer the relevance of both indices, CIAF and CIAF + anaemia. For instance, the “none” category of CIAF (Moran’s I = 0.632) is slightly more clustered than the “none” category of CIAF + anaemia index (Moran’s I = 0.547). On the other hand, the “only stunting” category of CIAF + anaemia (Moran’s I = 0.422) shows higher spatial autocorrelation compared to the “only stunting” category of CIAF (Moran’s I = 0.306). The nutrition failures or co-occurrences called “SUA”, “SWUA”, and “only anaemia” of the CIAF + anaemia index were found to moderately cluster across districts of India with positive spatial autocorrelation (Moran’s I = 0.588, 0.555, and 0.511, respectively), whereas the co-occurrence of stunting and anaemia (“SA”) had a comparatively lower magnitude of spatial clustering (Moran’s I = 0.285). All the values of Moran’s I discussed above were found to be significant in nature with a *p*-value less than 0.01.

Table 7 represents findings from spatial analysis performed in the NFHS-5 dataset to assess spatial autocorrelation between the existing forms of undernutrition and co-occurrences from the traditional CIAF and co-occurrences from the modified CIAF + anaemia across districts of India. The aim is to understand the spatial association between the general indicators or existing forms of undernutrition that are popularly used (like stunting, underweight, and anaemia) with those of the co-occurrences identified from indices like CIAF and CIAF + anaemia. Again, some selected combinations are analysed. As the “SUA” category from the CIAF + anaemia index was one of the most prevalent triple burdens of child undernutrition, we assess its spatial correlation with three existing forms of child undernutrition, namely, stunting, underweight, and anaemia. The positive value suggests a moderate positive spatial co-clustering between the co-occurrence of stunting, underweight, and anaemia (SUA) with all three indicators of stunting, underweight, and anaemia. Out of all three indicators, underweight was found to be the most clustered with the nutrition failure “SUA” (Moran’s I = 0.578), followed by the other two indicators of stunting (Moran’s I = 0.491) and anaemia (Moran’s I = 0.404). This means areas with high anaemia prevalence were more likely to be located near areas with high “SUA” prevalence, and regions with low anaemia prevalence were more likely to be near areas with low “SUA” prevalence. However, when we study the spatial dependence of the co-occurrence of stunting and anaemia “SA” with that of the existing forms of undernutrition, namely, stunting or anaemia, the association, although positive, was observed to be less strong than what was observed in the case of the co-occurrence of “SUA”. The co-occurrence “SA” was found to be more clustered with anaemia (Moran’s I = 0.238) than with stunting (Moran’s I = 0.232). On testing whether anaemia and “only anaemia” without the occurrence of stunting, wasting, and underweight co-occur spatially, a weak positive spatial relationship was observed across Indian districts (Moran’s I = 0.188). Similarly, the negative value of Moran’s I of −0.039 suggests a weak negative spatial relationship between the indicator of stunting and “only stunting” identified from CIAF + anaemia. This implies that areas with high stunting prevalence might not necessarily be located near areas with high “only stunting” prevalence of the CIAF + anaemia index, and vice versa. However, the positive value of Moran’s I of 0.16 indicated a weak positive spatial relationship between the “stunting” and “only stunting” categories from CIAF. This implies that areas with high stunting prevalence might be somewhat clustered near areas with high “only stunting” prevalence of the CIAF index and vice versa. All the values of Moran’s I discussed above were found to be significant in nature with a *p*-value less than 0.01.

### 3.5. Spatial Covariates of the Quadruple Burden of Child Undernutrition “SWUA” in India

Table 8 presents the magnitude of spatial autocorrelation for the district-level covariates taken into consideration for analysing the outcome variable, that is, the district-level prevalence of quadruple burden of child undernutrition using estimates from the NFHS-5 (2019–2021) dataset. While interpreting the findings from univariate local Moran’s I, most explanatory variables were found to be positively clustered across districts of India with Moran’s I above 0.4, except for a few indicators like the percentage of male children, the percentage of children who were sick in the last 2 weeks preceding the survey, and the percentage of children with less than 2 years of age. From the findings of bivariate local Moran’s I, the highest amount of co-clustering of the quadruple burden of child undernutrition was observed in the case of the indicator of underweight mothers (Moran’s I 0.511; *p*-value < 0.01), followed by the indicator of open defecation (Moran’s I 0.432; *p*-value < 0.01). To identify the hotspots and cold spots of the observed co-clustering with selected explanatory variables, bivariate LISA maps were prepared and can be accessed from Appendix A in the Appendix A.

Table 9 presents estimates from the spatial lag regression model for assessing the association between the quadruple burden of child undernutrition and different background, maternal, and child characteristics, NFHS-5 (2019–2021). Out of the seventeen covariates taken in the bivariate analysis, five variables were dropped from the regression analysis as they showed negligible or insignificant levels of co-clustering. The significant lag coefficient (rho = 0.468, *p* < 0.01) suggests that the quadruple burden of child undernutrition has strong spatial dependence among districts of India. Among background characteristics, the percentage of children in poorer households, open defecation, and lack of clean cooking fuel were positively and significantly associated with the quadruple burden of child undernutrition, highlighting poor living conditions as critical risk factors. It was found that if a district’s prevalence of children belonging to underweight mothers goes up by 10%, then there was a significant chance of an increase in the prevalence of the quadruple burden of child undernutrition by 0.82% (Coefficient = 0.082, *p* < 0.01).

## 4. Discussion

The present study began by analysing the prevalence and trends of the four child undernutrition outcomes: stunting, wasting, underweight, and anaemia. The study outlined the rising trends of anaemia, which is currently an alarming public health concern in India [16]. The study also analysed the age patterns of the four child undernutrition outcomes that portray a pattern similar to what was observed in the CNNS 2016–18 Report [17].

The study further proceeded to compute the prevalence of the traditional composite index of anthropometric failure as coined by Svedberg and Nandy [2,4]. The importance of studying CIAF is immense, as it is seen as a better indicator to assess undernutrition than the existing measures generally used, which are stunting, wasting, and underweight [18]. The literature highlights that areas with a high prevalence of co-occurrences of stunting, wasting, and underweight carry a higher burden of child mortality in India [19,20,21]. The findings of the present paper were in convergence with similar studies that had computed CIAF using datasets of NFHS-5 and CNNS, which found that around half of the children were free from stunting, wasting, and underweight, and the other half were burdened with one issue or the other [22,23,24].

The present study also tried to modify the CIAF by including anaemia as a substitute for the fourth form of undernutrition, which is micronutrient deficiencies, the first three being stunting, wasting, and underweight. This way, the newly formed index of CIAF + anaemia tried to encapsulate all four major forms of child undernutrition holistically. But this was not the first time one has attempted to modify the CIAF. Given the rising trend of overnutrition, many have tried to add overweight or obese as a component alongside stunting, wasting, and underweight into the CIAF [25,26,27]. Some have tried to modify CIAF by replacing underweight indicator calculated from weight-for-height with that computed from BMI-for-age [28]. Besides the CIAF, various studies in the literature have determined co-occurrences among different child malnutrition indicators [29,30,31,32].

The present study identified the dominant role of two child undernutrition indicators; one is stunting, which signifies chronic undernutrition and is well-known for its scarring effect; the other is anaemia, a silent killer of productivity and one’s well-being, which has recently caught everyone’s eyes, given its rising trend. There is a wide literature that analyses the co-occurrence of especially stunting and anaemia, as it is bound to have stark consequences for one’s health [12,33,34,35,36]. Analysing the co-occurrence of stunting and wasting is another popular phenomenon [37,38,39,40].

There is a dearth of research in India that studies the co-occurrence of all four outcomes as studied in this paper, but a study conducted in Ethiopia has performed a similar exercise and found that two-thirds of the children had at least one or the other issue out of the four undernutrition conditions, namely, stunting, wasting, underweight, and anaemia, and called this index the multiple nutrition deficits index [41]. Similar to what was observed in our study, it studied the association of multiple nutrition deficits and discovered that male children, those older, from poorer households, and with mothers belonging to the illiterate category were more likely to have multiple nutrition deficits.

Varghese (2019) analysed the mean prevalences of the dual burden of stunting, anaemia, and overweight among children at the individual and population levels with respect to states and districts, and, similar to our study, a strong presence of stunting and anaemia was also found at the district level [13]. As our research identified the worst-performing districts in terms of the quadruple burden of child undernutrition, similarly, another study using NFHS-4 data identified hotspots of higher prevalences of co-occurrences of stunting, wasting, and anaemia [42].

The country nutrition profiles maintained by the global nutrition report assessed that India is “on course” to achieve three of its global nutrition targets in 2025, “off course” for achieving targets on seven indicators, and “a worsening trend” for two indicators, which include wasting and anaemia [43]. The current scenario of child undernutrition in India calls for urgent actions to plan the effective use of nutrition-generated data for effective pathways for our country to meet its SDG 2030 targets [44].

The Indian government’s oldest flagship program to prevent undernutrition had been ICDS, launched in 1975, followed by mid-day meal in 1995. Since then, the government has brought several other schemes and programs like RBSK under NHM, the National Food Security Act, and the Swachh Bharat mission that help in fighting undernutrition challenges in India. The most recent initiatives include POSHAN Abhiyaan, Anaemia Mukt Bharat, and PMMVY. Despite the government’s efforts, there is a requirement for supportive supervision in each of these programs to ensure their effectiveness on the ground and develop a multidimensional approach to address the burden of malnutrition in India. There is a need for effective convergence strategies among different ministries to come together for a joint mission of a “suposhit Bharat” [45].

This study identifies mechanisms to identify nutritionally at-risk children and recommends catering to the needs of such children, prioritising those who are burdened with more than one form of undernutrition and those belonging to a nutritionally at-risk district. A system that tracks such families and highlights a scoring system based on these parameters can help identify who needs the most support. In conclusion, analysing such burdens with the help of smart data use can help to reduce the burden of undernutrition in our country and can be a supportive step for our government, as it has the potential for substantial programmatic implications [46]. There is a need to understand the co-morbidities of nutritional outcomes and gain more insights on multifactorial approaches for enhancing public health nutrition interventions accordingly [47]. Action can be directed towards investing in children with multiple nutritional deficits with enhanced tracking, screening, and service delivery across public health platforms.

This study tried to make a unique contribution to identifying the burdens of child undernutrition at individual and population levels, but it also carries some limitations. Firstly, there exists scope for improvement in the data quality of different nutrition indicators [48]. The data of any large-scale survey carry missing values, which do have a scope of imputation to strengthen the completeness of data, which has not been attempted in this study. The fact that NFHS-5 (2019–2021) was collected in two phases, as per the nation’s COVID restrictions, is bound to impact the quality of its data, as it also elongated the survey schedule. Secondly, there is an argument regarding the suitability of one-size-fits-all WHO standards of growth and other nutrition indicators in the Indian context, as misclassification of a child can also add to the inflation of child undernutrition outcomes [49]. Additionally, this study uses anaemia as a proxy for the fourth form of undernutrition, which is micronutrient deficiencies. But it is only 70% of the time when anaemia is caused due to nutritional deficiencies (iron-deficiency anaemia, folate- or B12-deficiency anaemia, or dimorphic anaemia), and the remaining 30% of the time, it could be due to anaemia of inflammation or anaemia of other causes [50]. Besides, in NFHS-5, anaemia was measured using capillary blood samples, which are more convenient for field surveys, though venous samples are generally more accurate. Thirdly, the study used data that were collected using a cross-sectional design, which makes it unfit for establishing causality and requires further in-depth research. Because this study is based on secondary data sources, it is bound to miss capturing the variation caused by indicators whose data were not collected in the survey. To cover information on such chance causes, this study involved statistical calculations, whose findings are subject to the fulfilment of typical assumptions that may be violated in specific scenarios.

The study also offers a further scope of research in addressing child undernutrition in India. There is scope for state-specific scrutiny to dig deeper and analyse the factors playing a role in state-specific contexts. There is scope for future research for getting more robust scientific evidence to confirm whether the co-occurrences like “SUA”, “SA”, and “only anaemia” play a distinct role from the previously used indicators of stunting, wasting, underweight, and anaemia, requiring specialised treatments. As all four forms of undernutrition co-occur and often have the same population risk factors, there is a chance of a bi-directional or multi-directional time-dependent relationship among the four forms of child undernutrition. There is a need for evidence and inference to understand how these four forms of undernutrition co-occur dynamically, what changes they cause biologically, and how they cause or interact with one another.

## 5. Conclusions

This paper tried to study the co-occurrences of stunting, wasting, underweight, and anaemia under an index called “CIAF + anaemia” among under-five children in India using NFHS data. The new index identified the co-occurrences, namely, “SUA” and “SA”, as being the most prevalent triple burden and dual burden of child undernutrition, respectively, which were also found to be moderately and positively clustered at the population level across the districts of India. Hence, it can be concluded that studying the co-occurrences does offer additional insights into the existing forms of child undernutrition. This research studied the spatial pattern of the quadruple burden of child undernutrition and identified the worst-performing pockets in the states of Jharkhand and Maharashtra. The bivariate analysis inferred that the co-occurrence “SUA” was more prevalent among children having undernourished mothers, whereas that of the standalone condition “only anaemia” was found more among children with overweight or obese mothers. In most scenarios, the condition of “only anaemia” showed an opposite pattern compared to the pattern observed in the case of the dual and triple burdens of child undernutrition. From the spatial analysis, it was found that the indicator of underweight mothers was one of the most important predictors of the quadruple burden of child undernutrition in India. Overall, the present study contributed to understanding the co-occurrences, burdens, and clustering of child undernutrition at the individual level as well as the population level. These can serve as monitoring measures for the government of India in identifying such high-risk individuals or districts and targeting interventions for marginalised groups accordingly.

## Figures and Tables

**Figure 1 nutrients-17-00977-f001:**
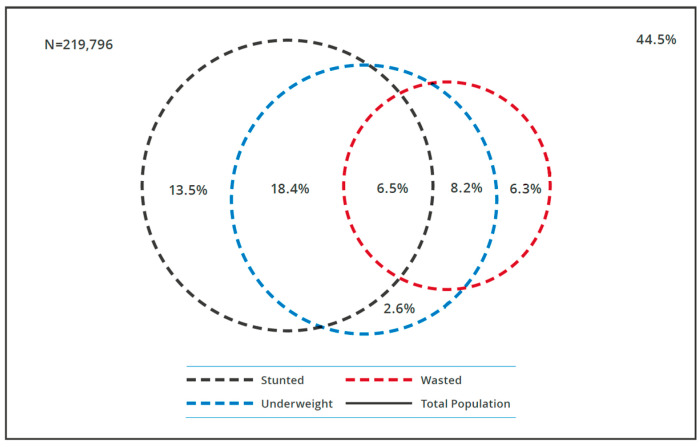
Multiple forms of child undernutrition, India, NFHS 2015–2016. Source: Food and Nutrition Security Analysis, a report by MOSPI, Government of India, 2019.

**Figure 2 nutrients-17-00977-f002:**
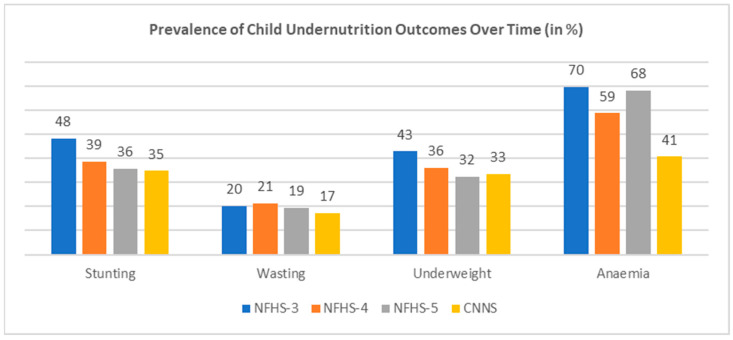
Trend of existing indicators of forms of child undernutrition (stunting, wasting, underweight, and anaemia), NFHS-3 (2005–2006), NFHS-4 (2015–2016), NFHS-5 (2019–2021), and the CNNS (2016–2018).

**Figure 3 nutrients-17-00977-f003:**
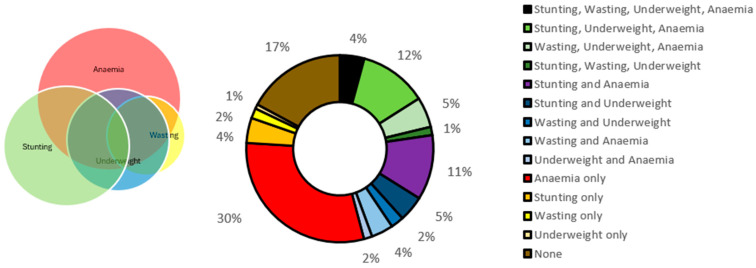
Distribution of the outcome variable “modified CIAF+ anaemia” comprising the co-occurrences of stunting (S), wasting (W), underweight (U), and child anaemia (A) among 6–59 month old children, NFHS-5 (2019–2021).

**Figure 4 nutrients-17-00977-f004:**
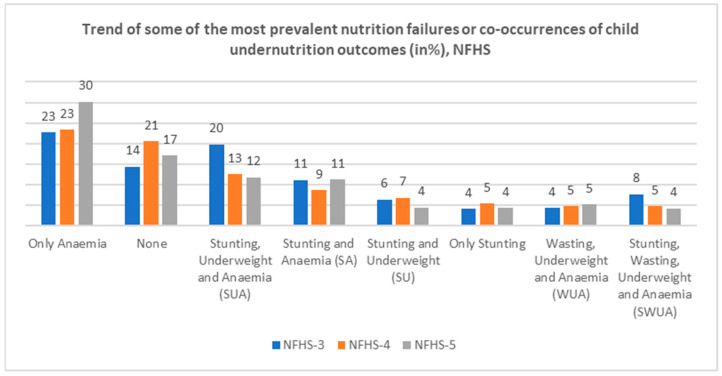
Trend in the prevalence (in %) of some of the most prevalent co-occurrences of undernutrition outcomes discovered from the modified index of CIAF + anaemia across NFHS-3 (2005–2006), NFHS-4 (2015–2016), and NFHS-5 (2019–2021).

**Figure 5 nutrients-17-00977-f005:**
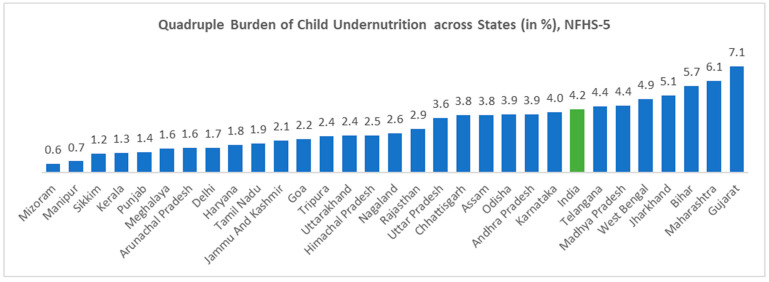
Prevalence of the quadruple burden of undernutrition, that is, the prevalence of the co-occurrence of all four forms of undernutrition, that is, stunting, wasting, underweight, and anaemia (SWUA) in children aged 6–59 months across the states of India (in %), NFHS-5 2019–2021.

**Figure 6 nutrients-17-00977-f006:**
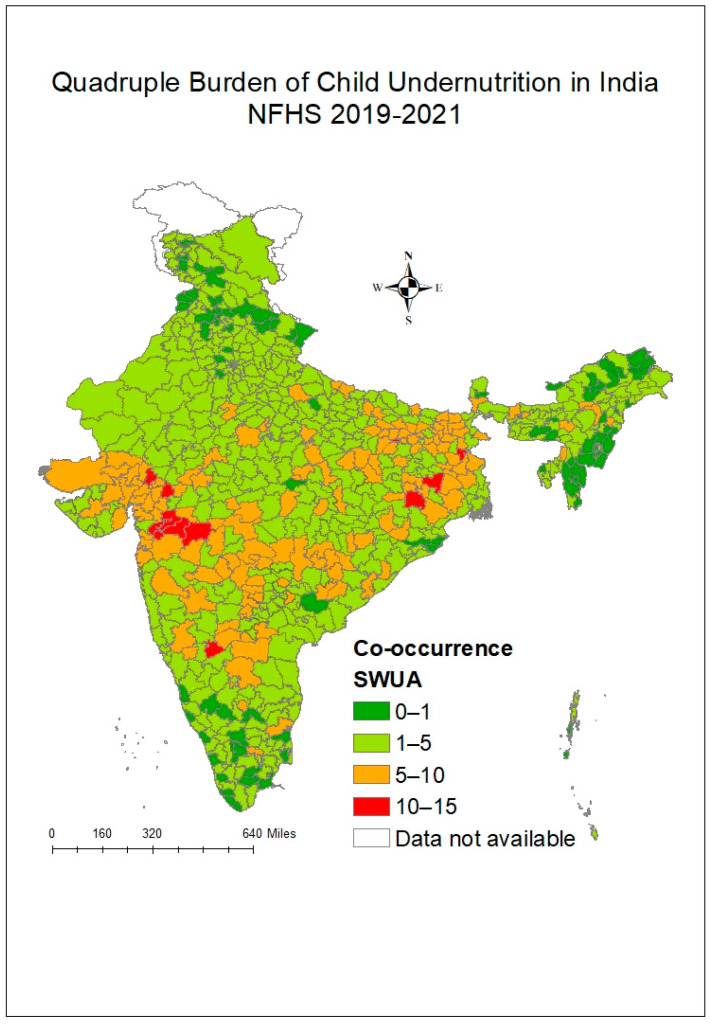
Quadruple burden (in %) across the districts of India, NFHS-5 2019–2021.

**Figure 7 nutrients-17-00977-f007:**
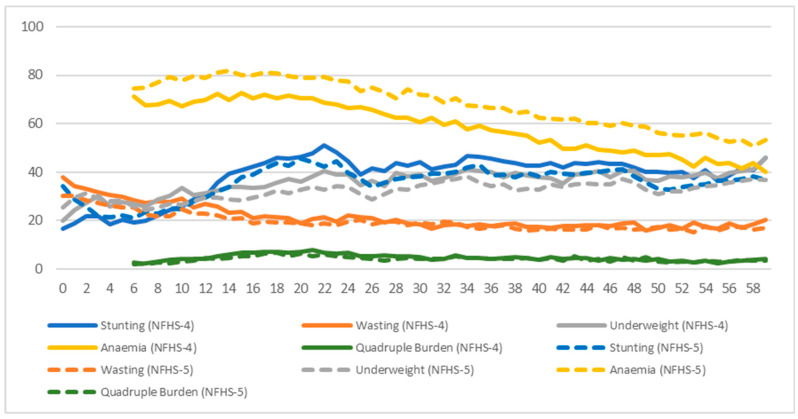
Age pattern of stunting, wasting, underweight status, anaemia, and quadruple burden (in %) with respect to NFHS-4 (2015–16) and NFHS-5 (2019–21).

**Table 1 nutrients-17-00977-t001:** Mutually exclusive categories of the outcome variable, modified CIAF + anaemia.

Type of Burdens	Co-Occurrences or Nutrition Failures (NFs)	
**Quadruple Burden of Child Undernutrition**	Co-occurrence of Stunting, Wasting, Underweight, Anaemia	SWUA
**Triple Burdens of Child Undernutrition**	Co-occurrence of Stunting, Underweight, Anaemia	SUA
	Co-occurrence of Wasting, Underweight, Anaemia	WUA
	Co-occurrence of Stunting, Wasting, Underweight	SWU
**Dual Burdens of Child Undernutrition**	Co-occurrence of Stunting and Anaemia	SA
	Co-occurrence of Stunting and Underweight	SU
	Co-occurrence of Wasting and Underweight	WU
	Co-occurrence of Wasting and Anaemia	WA
	Co-occurrence of Underweight and Anaemia	UA
**Single Burdens of Child Undernutrition (Standalone Forms)**	Occurrence of only Anaemia	Only Anaemia
	Occurrence of only Stunting	Only Stunting
	Occurrence of only Wasting	Only Wasting
	Occurrence of only Underweight	Only Underweight
**No Burden**	None (Free from child undernutrition)	None

**Table 2 nutrients-17-00977-t002:** Description and coding of each explanatory variable used, NFHS.

Variable	Variable Description
**Region**	Indian states are divided into six regions, namely, North, Central, East, Northeast, West, and South.
**EAG State**	It is a grouping variable that categorised the states into two categories, “EAG” and “Non-EAG”. The Empowered Action Group (EAG) states are a group of eight states comprising Bihar, Chhattisgarh, Jharkhand, Madhya Pradesh, Odisha, Rajasthan, Uttarakhand, and Uttar Pradesh. The Government of India established the group in 2001 to support the development of area-specific programs, as these regions struggled to control population growth at a manageable level.
**Place of Residence**	India is divided into urban and rural clusters in the NFHS dataset.
**Social Group**	India’s social structure comprises different communities, castes, and tribes, which are generally categorised as “scheduled castes”, “scheduled tribes”, “other backward class”, and “other”.
**Wealth Index of the Household**	As per DHS methodology, wealth index scores computed using principal component analysis (PCA) are generated for each household based on its amenities, and this variable serves as a proxy for analysing the socioeconomic status of the household. It originally comprised five categories: poor, poorer, middle, richer, and richest in the NFHS dataset. In this variable, the first two categories were clubbed into poorer and the last two into richer, for the final variable to have three categories, namely, “poorer”, “middle”, and “richer”.
**Toilet Facilities in Household**	The NFHS household tool asks the head of the household whether they have access to a toilet or not; if yes, then which one, where is it located, and whether it is shared. As per the NFHS, improved sanitation facilities refer to any non-shared toilet of the following types: flush/pour flush toilets to piped sewer systems, septic tanks, pit latrines, or an unknown destination; ventilated improved pit (VIP)/biogas latrines; pit latrines with slabs; and twin pit/composting toilets. The final variable was coded into three categories: improved, unimproved, and open defecation.
**Education of Mother**	The education of mothers was divided into two categories, “illiterate” and “literate”. Illiterate refers to the fact that the mother never went to school.
**Maternal Height Categorised**	Maternal height was divided into three categories: short, normal, tall, as well as not measured. Height less than 145 cm was defined as short, height 160 cm and above was defined as tall, and otherwise as normal.
**Body Mass Index**	As per the literature, the body mass index (BMI) of an adult is categorised as “underweight” if it is less than 18.5, “overweight or obese” if it is more than 25, and otherwise as “normal”. A separate category was created for pregnant women or missing values.
**Diet of Mother**	The NFHS women’s questionnaire asks women how often they eat seven types of food items (milk or curd; pulses or beans; dark green leafy vegetables; fruits; eggs; fish; chicken or meat), with responses including 0 = never; 1 = occasionally; 2 = weekly; 3 = daily. Responses for all 7 items were added, and an additive score was generated ranging from 0 to 21; The score was divided into three categories to serve as a proxy for ranking the general dietary intake of the mother; scores less than 11 meant a diet “low” in intake, scores from 11 to 14 meant a “medium” intake diet, and scores of 15 or more were defined as a “high” intake diet.
**Mother’s Age at Childbirth**	The mother’s age at childbirth was categorised into <19; 19–25, 25–34, 35–44, and 45+ years of age.
**“Wantedness” of Child**	The mother was asked, regarding her youngest child under the age of five, if she “wanted” to have a child ever or later when she came to know about her pregnancy.
**Institutional Delivery**	The mother was asked if she delivered her child under the age of five at a health facility.
**Birth Order**	Birth order refers to the sequence in which children are born to a family. It was divided into three categories: “1 bo”, “2–3 bo”, and “4 or more”.
**Birth Weight**	“Low birth weight” refers to a child who weighed 2.5 kg or less at the time of birth, “high birth weight” refers to 4.5 kg or more, and otherwise as “normal”.
**Sex of the Child**	Sex was categorised into two categories, “male” and “female”.
**Child Age**	Child age was categorised into either less than 2 years of age or above.

**Table 3 nutrients-17-00977-t003:** The prevalence of co-occurrences (in %) of child undernutrition outcomes as part of the traditional CIAF and the modified CIAF + anaemia, NFHS-5 (2019–2021).

Type of Burdens	Co-Occurrences	Prevalence of Co-Occurrences	Prevalence of Burdens
**The co-occurrences from the traditional CIAF**			
**All three**	Stunting, Wasting, Underweight	5.3	5.3
**Dual burdens**	Stunting and Underweight	15.7	23.3
	Wasting and Underweight	7.6	
**Single burdens (or standalone forms)**	Stunting only	15.2	23.8
	Wasting only	6.4	
	Underweight only	2.3	
**No burden**	None	47.6	47.6
**The co-occurrences from the modified CIAF + anaemia**			
**All four**	Stunting, Wasting, Underweight, Anaemia (SWUA)	4.2	4.2
**Triple burdens**	Stunting, Underweight, Anaemia (SUA)	11.8	18.4
	Wasting, Underweight, Anaemia (WUA)	5.2	
	Stunting, Wasting, Underweight (SWU)	1.4	
**Dual burdens**	Stunting and Anaemia (SA)	11.1	23.1
	Stunting and Underweight (SU)	4.5	
	Wasting and Underweight (WU)	2.2	
	Wasting and Anaemia (WA)	3.8	
	Underweight and Anaemia (UA)	1.5	
**Single burden (or standalone forms)**	Anaemia only	30.3	37.1
	Stunting only	4.3	
	Wasting only	1.7	
	Underweight only	0.8	
**No burden**	None	17.2	17.2

**Table 4 nutrients-17-00977-t004:** Prevalence of different co-occurrences of child undernutrition (in %) discovered from the newly created index of modified CIAF + anaemia with respect to background characteristics and maternal and child characteristics, NFHS 2019–2021.

Prevalence (in %)	No Burden	Quadruple Burden	Triple Burdens			Dual Burdens					Single Burdens (or Standalone Forms)			
	None	SWUA	WUA	SWU	SUA	SA	SU	WU	WA	UA	Only Anaemia	Only Stunting	Only Wasting	Only Underweight
**Region**														
North	18.8	2.4	4.0	0.8	9.1	11.0	3.1	1.4	3.5	1.2	38.8	3.9	1.5	0.5
Central	15.9	3.8	4.6	1.3	12.6	12.9	5.0	2.2	3.4	1.3	29.7	5.0	1.5	0.8
East	15.3	5.2	6.5	1.7	13.8	10.7	5.3	2.3	3.8	1.7	27.3	4.0	1.6	0.8
Northeast	19.1	3.2	5.1	1.4	10.1	11.4	4.4	2.4	4.6	1.3	28.2	5.8	2.4	0.8
West	13.2	6.4	7.2	1.6	12.7	10.8	3.7	2.7	5.1	2.0	28.9	3.1	1.8	0.8
South	24.0	3.1	3.8	1.3	9.0	9.1	4.0	2.5	3.3	1.5	30.8	4.5	2.3	1.0
**Empowered Action Group (EAG) State**														
Non-EAG	19.1	4.1	5.1	1.3	10.2	10.2	3.8	2.4	4.0	1.6	31.4	4.1	1.9	0.8
EAG	15.6	4.2	5.3	1.5	13.1	11.9	5.1	2.1	3.5	1.4	29.3	4.5	1.6	0.8
**Place of residence**														
Urban	20.6	3.2	4.5	1.3	9.1	9.9	3.6	2.4	4.2	1.3	32.5	4.3	2.3	0.7
Rural	16.0	4.5	5.5	1.4	12.7	11.5	4.8	2.2	3.6	1.6	29.5	4.3	1.5	0.8
**Social Group**														
Scheduled Caste (SC)	14.9	4.7	5.5	1.4	14.2	11.9	4.7	2.0	3.6	1.6	28.8	4.4	1.4	0.7
Scheduled Tribe (ST)	12.6	6.4	7.2	1.4	15.1	11.5	4.4	2.0	4.3	1.8	27.5	3.7	1.6	0.7
Other Backward Class (OBC)	18.1	4.0	4.9	1.5	11.2	11.0	4.7	2.4	3.6	1.5	29.9	4.4	1.9	0.8
Others	19.9	3.1	4.7	1.0	8.9	10.3	3.8	2.2	4.0	1.4	33.8	4.2	1.9	0.8
**Wealth Index of the Household**														
Poorer	12.9	5.6	6.2	1.7	15.8	12.4	5.5	2.2	3.6	1.7	25.6	4.4	1.4	0.8
Middle	17.3	3.8	4.7	1.2	10.7	11.1	4.6	2.5	3.6	1.5	32.0	4.5	1.7	0.7
Richer	23.1	2.4	4.2	1.1	6.8	9.4	2.9	2.2	4.0	1.3	35.7	4.0	2.2	0.8
**Toilet Facility in Household**														
Improved, not shared	19.5	3.4	4.9	1.3	9.8	10.4	4.0	2.3	3.9	1.4	32.1	4.4	2.0	0.8
Unimproved source/Shared/Other	16.2	4.0	5.3	1.4	11.7	11.7	4.4	2.3	3.8	1.5	31.1	4.2	1.5	0.9
Open defecation	12.1	6.3	6.1	1.7	17.0	12.6	5.7	2.1	3.4	1.7	25.0	4.1	1.3	0.7
**Maternal and Child Characteristics**														
**Education of Mother**														
Illiterate	11.5	6.0	6.3	1.7	17.9	13.0	5.8	2.0	3.6	1.5	24.2	4.3	1.4	0.7
Literate	18.8	3.7	4.9	1.3	10.1	10.6	4.1	2.3	3.8	1.5	32.0	4.3	1.8	0.8
**Maternal Height Categorised**														
Short	10.4	6.7	5.4	2.2	20.3	13.6	8.1	2.0	2.8	1.5	19.3	5.7	1.3	0.8
Normal	17.6	4.0	5.3	1.3	11.1	11.0	4.2	2.3	3.9	1.5	30.9	4.2	1.7	0.8
Tall	23.3	2.2	4.1	0.7	5.6	8.0	2.0	2.0	4.2	1.3	40.7	3.1	2.2	0.6
Not measured/Missing	15.6	5.5	6.3	1.3	12.6	16.5	3.0	2.5	4.3	0.4	25.4	3.9	2.2	0.5
**Body Mass Index of Mother**														
Underweight	11.7	7.2	7.2	2.1	16.1	11.0	5.2	2.4	3.7	2.2	25.3	3.6	1.5	1.0
Normal	16.8	3.7	5.2	1.3	11.5	11.2	4.5	2.4	4.2	1.4	30.7	4.4	1.9	0.7
Overweight	25.4	1.9	3.1	0.8	7.0	9.9	3.4	1.7	2.9	1.0	35.5	4.8	1.6	0.9
Missing/Currently Pregnant	14.9	5.0	5.1	1.5	14.7	13.0	5.0	1.9	2.9	1.6	27.9	4.4	1.4	0.7
**Diet of Mother**														
Low	15.6	4.4	5.2	1.3	12.5	12.0	4.5	2.1	3.6	1.6	30.9	4.1	1.4	0.8
Medium	17.0	4.3	5.4	1.5	12.1	10.9	4.7	2.2	3.7	1.5	29.7	4.4	1.8	0.8
High	20.2	3.5	5.0	1.3	10.2	10.3	3.9	2.4	4.0	1.4	30.5	4.4	2.1	0.9
**Mother’s Age at Childbirth**														
<19	15.5	5.5	5.6	1.5	14.3	11.0	5.2	2.3	3.7	1.7	27.2	4.1	1.3	1.0
19–25	17.1	4.2	5.1	1.5	11.8	11.2	4.5	2.2	3.7	1.6	30.3	4.3	1.7	0.8
25–34	17.8	3.9	5.3	1.2	11.1	10.9	4.2	2.3	3.9	1.3	31.4	4.2	1.8	0.7
35–44	18.3	4.3	5.5	1.4	11.7	11.1	4.6	2.1	3.8	1.5	27.2	4.7	2.8	0.8
45+	8.0	6.3	25.9	2.2	12.6	13.3	3.2	0.6	6.2	0.1	17.0	1.6	3.1	0.1
**“Wantedness” of Child**														
No More/Later	15.3	4.7	5.0	1.8	13.9	11.8	4.6	1.8	3.7	1.9	29.8	3.9	1.1	0.7
Wanted	17.4	4.1	5.2	1.4	11.6	11.1	4.4	2.3	3.8	1.5	30.3	4.3	1.8	0.8
**Institutional Delivery of Child**														
No	13.1	5.7	5.5	2.0	16.9	12.9	6.3	2.2	3.3	1.8	23.7	4.7	1.3	0.8
Yes	17.8	4.0	5.2	1.3	11.1	10.9	4.2	2.2	3.8	1.5	31.2	4.2	1.8	0.8
**Birth Order of the Child**														
1 bo	19.8	3.5	4.8	1.2	9.4	10.3	3.9	2.3	3.9	1.5	32.7	4.3	1.8	0.8
2–3 bo	16.5	4.3	5.5	1.5	12.3	11.2	4.6	2.2	3.7	1.6	29.8	4.2	1.8	0.8
4 or more	12.1	5.7	5.7	1.7	17.3	13.5	5.6	2.1	3.4	1.5	24.6	4.6	1.5	0.7
**Birth Weight of the Child**														
Low	13.6	6.3	6.3	2.2	15.8	11.1	5.7	2.6	3.7	1.7	24.6	4.0	1.4	1.0
Normal	18.0	3.7	5.0	1.2	10.9	11.1	4.2	2.2	3.8	1.5	31.6	4.4	1.8	0.8
High	17.1	4.5	5.5	1.6	12.8	11.3	5.4	2.1	3.7	1.2	29.3	4.0	1.1	0.5
Not weighed/Not known	13.2	5.7	6.2	1.9	16.1	13.2	5.9	1.6	3.6	1.4	24.4	4.8	1.4	0.7
**Sex of the Child**														
Male	17.1	4.7	5.4	1.5	11.6	11.4	4.4	2.3	3.7	1.4	29.8	4.4	1.7	0.7
Female	17.3	3.6	5.1	1.3	12.0	10.9	4.5	2.1	3.8	1.6	30.9	4.2	1.8	0.9
**Child’s Age**														
6–23 months	11.1	4.6	6.7	0.9	10.2	14.0	2.1	1.7	5.4	1.4	36.9	3.2	1.5	0.3
24–59 months	19.9	4.0	4.6	1.6	12.5	9.9	5.5	2.5	3.0	1.6	27.4	4.8	1.8	1.0

Note: The association between the outcome variable of CIAF + anaemia consisting of the fourteen categories and the various independent variables mentioned in the table was tested using the chi-square test of association, and the *p*-value was found to be statistically significant at 1% for each of the independent variables (*p* < 0.001).

**Table 5 nutrients-17-00977-t005:** Estimates of the adjusted odds ratio and a 95% confidence interval from the ordered logistic regression results run for identifying the determinants of the ordered variables representing zero, single, dual, triple, or quadruple burdens of child undernutrition in India, NFHS-5 2019–2021.

	Adjusted Odds Ratio (AOR)	95% Confidence Interval		Adjusted Odds Ratio (AOR)	95% Confidence Interval
** Background Characteristics **			**Body Mass Index**		
**Region**			Normal ^®^	1.00	
North ^®^	1.00		Underweight	1.46	(1.41–1.52) ***
Central	1.07	(1.02–1.12) ***	Overweight	0.69	(0.66–0.71) ***
East	1.11	(1.05–1.16) ***	Missing/Currently Pregnant	1.23	(1.17–1.29) ***
Northeast	0.92	(0.86–0.97) ***	**Dietary Intake of Mother**		
West	1.68	(1.57–1.8) ***	Low ^®^	1.00	
South	0.97	(0.92–1.03) ^ns^	Medium	0.99	(0.97–1.03) ^ns^
**Empowered Action Group (EAG) State**			High	0.95	(0.92–0.99) **
Non-EAG State ^®^	1.00		**Mother’s Age at Childbirth**		
EAG State	0.97	(0.92–1.02) ^ns^	<19 ^®^	1.00	
**Place of Residence**			19–25	0.84	(0.79–0.89) ***
Urban ^®^	1.00		25–34	0.77	(0.72–0.82) ***
Rural	0.95	(0.91–0.99) **	35–44	0.70	(0.64–0.77) ***
**Social Group**			45+	1.65	(0.79–3.46) ^ns^
Scheduled Caste (SC)^ ®^	1.00		**“Wantedness” of Child**		
Scheduled Tribe (ST)	1.03	(0.99–1.09) ^ns^	No more/Later ^®^	1.00	
Other Backward Class (OBC)	0.91	(0.88–0.94) ***	Wanted	0.98	(0.94–1.04) ^ns^
General/Others	0.82	(0.78–0.85) ***	**Institutional Delivery**		
**Wealth Index of the household**			No ^®^	1.00	
Poorer ^®^	1.00		Yes	0.90	(0.87–0.94) ***
Middle	0.83	(0.8–0.86) ***	**Birth Order**		
Richer	0.67	(0.65–0.71) ***	1 bo ^®^	1.00	
**Toilet Facility in Household**			2–3 bo	1.29	(1.25–1.32) ***
Improved, Not Shared ^®^	1.00		4 or more	1.53	(1.46–1.61) ***
Unimproved Source/Shared/Other	1.04	(1–1.08) **	**Birth Weight**		
Open Defecation	1.14	(1.1–1.18) ***	Low ^®^	1.00	
** Maternal and Child Characteristics **			Normal	0.67	(0.64–0.69) ***
**Education of Mother**			High	0.64	(0.59–0.7) ***
Illiterate ^®^	1.00		Not weighed/Not known	0.69	(0.61–0.79) ***
Literate	0.77	(0.74–0.79) ***	**Sex of the Child**		
**Maternal Height Categorised**			Male ^®^	1.00	
Short ^®^	1.00		female	0.92	(0.9–0.94) ***
Normal	0.57	(0.55–0.6) ***	**Child Age**		
Tall	0.40	(0.38–0.42) ***	6–23 months ^®^	1.00	
Not Measured/Missing	0.91	(0.41–2.02) ^ns^	24–59 months	0.82	(0.8–0.85) ***
**Estimated Cut Points**					
Cut 1	−3.54	(−4.66, −2.42)	Cut 3	−0.52	(−1.64, 0.60)
Cut 2	−1.67	(−2.79, −0.55)	Cut 4	1.46	(0.35, 2.58)

Note: ^®^ represents reference category; *** represents *p* < 0.01, ** represents *p* < 0.05, ns represents not significant.

**Table 6 nutrients-17-00977-t006:** Univariate Moran’s I computed for assessing clustering of existing forms of undernutrition, co-occurrences from the traditional CIAF, and co-occurrences from the modified CIAF + anaemia version across districts in India, NFHS-5 (2019–2021).

Indicators	Univariate Moran’s I
**Child undernutrition outcomes**	
Stunting	0.511
Wasting	0.463
Underweight	0.677
Anaemia	0.579
**Co-occurrences from the index CIAF + anaemia**	
Co-occurrence of Stunting, Underweight, and Anaemia (SUA)	0.588
Co-occurrence of Stunting and Anaemia (SA)	0.285
Co-occurrence of Stunting, Wasting, Underweight, and Anaemia (SWUA)	0.555
Stunting, without co-occurrence of any other undernutrition issue like wasting, underweight, or anaemia (only stunting)	0.422
Anaemia, without co-occurrence of any other undernutrition issue like stunting, wasting, underweight (only anaemia)	0.511
Free from any form of undernutrition like stunting, wasting, underweight, or anaemia (none)	0.547
**Co-occurrences from the index CIAF**	
Stunting, without co-occurrence of any other undernutrition issue like wasting or underweight (only stunting)	0.306
Free from any form of undernutrition like stunting, wasting, or underweight (none)	0.632

Note: All the above univariate local Moran’s I values were tested for significance based on analytical approximation and were found to be significant (*p* = 0.001). The pseudo-*p*-value was computed using random permutations in the GeoDa software v1.18.

**Table 7 nutrients-17-00977-t007:** Bivariate Moran’s I computed for assessing co-clustering in between existing forms of undernutrition and co-occurrences from the traditional CIAF and co-occurrences from the modified CIAF + anaemia, across districts of India, NFHS-5 (2019–2021).

Indicator-1 (Lagged)	Indicator-2	Bivariate Moran’s I	*p*-Value
**Child Undernutrition Indicators**	**Co-occurrences from CIAF + anaemia**		
Stunting	SUA	0.491	0.001
Underweight	SUA	0.578	0.001
Anaemia	SUA	0.404	0.001
Stunting	SA	0.232	0.001
Anaemia	SA	0.238	0.001
Anaemia	Only Anaemia	0.188	0.001
Stunting	Only Stunting	−0.039	0.006
**Child Undernutrition Indicators**	**Co-occurrences from CIAF**		
Stunting	Only Stunting	0.16	0.001

Note: The bivariate local Moran’s I values were tested for significance based on analytical approximation. The *p*-value represents the pseudo-*p*-value that was computed using random permutations in the GeoDa software version 1.18.

**Table 8 nutrients-17-00977-t008:** Estimates of spatial autocorrelation: (i) univariate local Moran’s I statistic for assessing clustering of explanatory variables and (ii) bivariate local Moran’s I statistic for assessing co-clustering of different explanatory variables with the outcome variable quadruple burden of child undernutrition.

District-Level Covariates	Univariate Local Moran’s I	*p*-Value	Bivariate Local Moran’s I	*p*-Value
**Background Characteristics**				
% Children residing in rural clusters	0.441	0.001	0.116	0.001
% Children belonging to poorer category households	0.749	0.001	0.276	0.001
% Children whose household practices open defecation	0.709	0.001	0.432	0.001
% Children whose household uses a clean cooking fuel	0.689	0.001	−0.206	0.001
% Children belonging to SC or ST social groups	0.616	0.001	0.001	0.463
**Maternal and Child Characteristics**				
% Children whose mothers were illiterate	0.681	0.001	0.302	0.001
% Children whose mothers had short height	0.654	0.001	0.283	0.001
% Children with underweight mothers	0.677	0.001	0.511	0.001
% Children with their mothers consuming a poor diet	0.844	0.001	0.09	0.001
% Children whose mother’s age at the time of their birth was below 18	0.693	0.001	0.268	0.001
% Children who were unwanted as per mothers	0.473	0.001	−0.04	0.012
% Children who had institutional delivery	0.66	0.001	−0.053	0.003
% Children born with low birth weight	0.541	0.001	0.168	0.001
% Children with birth order of 4 or more	0.659	0.001	0.145	0.001
% Male children	0.072	0.002	0.011	0.251
% Children of age 6–23 months	0.153	0.001	0.116	0.001
% Children who were sick in the last 2 weeks preceding the survey	0.383	0.001	0.091	0.001

Note: In the calculation of bivariate local Moran’s I, the covariates were taken as the lagged variable.

**Table 9 nutrients-17-00977-t009:** Spatial regression estimates of the quadruple burden of child undernutrition and its association with background, maternal, and child characteristics, NFHS-5 (2019–2021).

District-Level Covariates	Regression Coefficient (Standard Error)
Lag Coefficient (Rho)	0.468 ***
Constant Term	−1.372 *
**Background Characteristics**	
% Children residing in rural clusters	0.0001 (0.004) ^ns^
% Children belonging to poorer category households	0.010 (0.005) **
% Children whose household practices open defecation	0.011 (0.005) **
% Children from households using clean cooking fuel	0.009 (0.004) **
**Maternal and Child Characteristics**	
% Children whose mothers were illiterate	0.010 (0.006) ^ns^
% Children whose mothers had short height	−0.001 (0.014) ^ns^
% Children with underweight mothers	0.082 (0.011) ***
% Children with mother’s age at their birth below 18	0.019 (0.017) ^ns^
% Children born with low birth weight	0.022 (0.013) *
% Children with a birth order of 4 or more	0.004 (0.012) ^ns^
% Children of age 6–23 months	−0.003 (0.017) ^ns^
**Regression Diagnostic Parameters**	
R-square	0.465
Akaike info criterion (AIC)	2678.5

Note: *** represents *p* < 0.01, ** represents *p* < 0.05, * represents *p* < 0.1. ns represents not significant.

## Data Availability

Data supporting reported results can be accessed from www.dhsprogram.com, accessed on 19 March 2023.

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
