# Peer review of "Co-Occurrences of Forms of Child Undernutrition in India: Insights from the National Family Health Survey"

_nutrients, 2025, doi:10.3390/nu17060977_

Round 1
Reviewer 1 Report
Comments and Suggestions for Authors
Very kind,
thank you for your work, which I think is very interesting and important for the magazine.
Before carrying out a careful review, I would ask you to fix the presentation of the data.
1- box 1 in the introduction should be removed and replaced by a descriptive paragraph.
2- if a concept of significance is marked it should be followed by the p value e.g. A significant in line 167
3- Figure 3.2: Prevalence of CIAF, NFHS 2019-21 can be removed and replaced with a descriptive paragraph.
4- Figure 3.4: Trend of different burdens over time and survey rounds, NFHS and CNNS. seems complex, it should be redone more clearly, e.g. by inserting column other
5- Figure 3.5: Quadruple burden of Undernutrition, CNNS and NFHS-5 is superfluous and unclear
6- Table 3.2: Sample Description and Prevalence of different burdens of Child Undernutrition, this table is very chaotic, the numbers should be on the same line, it is recommended to split it into two tables! or to delete the unnecessary ones.
Thank you, I apologise for the revision, but for the readers of the article these are important elements
Author Response
S.no. |
Comment |
Response |
|
Very kind, thank you for your work, which I think is very interesting and important for the magazine. Before carrying out a careful review, I would ask you to fix the presentation of the data. |
Dear Reviewer, thank you very much for your time and feedback. Thank you for identifying areas of improvement in the manuscript. |
1 |
box 1 in the introduction should be removed and replaced by a descriptive paragraph. |
Agree. Explanation through a paragraph seems more appropriate. It is modified in the line 38 to 49 of the revised manuscript. |
2 |
if a concept of significance is marked it should be followed by the p value e.g. A significant in line 167 |
Correct. The word “significant” is incorrectly used here as no statistical test was involved. The statement just talks about the observed trend in the prevalence of stunting and underweight. The word “significant” has been removed now line 241 of the revised manuscript. |
3 |
Figure 3.2: Prevalence of CIAF, NFHS 2019-21 can be removed and replaced with a descriptive paragraph. |
Agree. Explanation through a paragraph exists in the text too. But I believe that it is important to highlight through a table on what was the distribution of the traditional CIAF index and how it differs from that of the modified index CIAF+Anemia. Hence, for ease of understanding, we have kept the distribution of the old and the modified index in a table (Table 3) together for comparison purposes. Explanation of the new index starts from line 266 of the revised manuscript. |
4 |
Figure 3.4: Trend of different burdens over time and survey rounds, NFHS and CNNS. seems complex, it should be redone more clearly, e.g. by inserting column other |
Thank you for your comment. For ease of understanding, we have now kept only the eight most prevalent burdens instead of all the fourteen categories of the new index of CIAF+Anemia in the Figure 4. To reduce complexity, we have also stuck to using only the three rounds of NFHS to show the trend and removed the prevalences of the CNNS survey. |
5 |
Figure 3.5: Quadruple burden of Undernutrition, CNNS and NFHS-5 is superfluous and unclear |
Thank you for your feedback. We have now explained it more thoroughly in the chart title, in the figure title and in the main text. It is the prevalence of one of the most severe conditions observed, as it is the percentage of those children who are suffering from all four forms of child undernutrition, that is, stunting, wasting, underweight and anemia. The figure 5 shows its prevalence across all the Indian states which gives us a geography wise perspective on the severity of the problem. To reduce complexity, I have only kept the NFHS as a data source and removed the prevalence of the CNNS survey. We have added the bar of “India” as well which was earlier not there. Now, it gives a comprehensive picture of the situation of the quadruple burden of child Undernutrition across Indian States as per the latest survey estimates of NFHS-5. |
6 |
Table 3.2: Sample Description and Prevalence of different burdens of Child Undernutrition, this table is very chaotic, the numbers should be on the same line, it is recommended to split it into two tables! or to delete the unnecessary ones. |
Agree. we have split it into two tables now, one for Sample description and the other for the Prevalence of different co-occurences of Child Undernutrition across background characteristics. We have shifted the table of Sample description to the supplementary material as Table S1. |
|
Thank you, I apologise for the revision, but for the readers of the article these are important elements |
Overall, we have worked on improving all the sections of the manuscript. Thank you so much for your feedback. |
Reviewer 2 Report
Comments and Suggestions for Authors
This study aimed to attempt to modify it by inclusion of a fourth form of undernutrition, that is anemia, serving as a proxy for micronutrient deficiencies among under-five children of India. As the results, the modified index of “composite Index of anthropometric failure (CIAF) + Anemia” identified thirteen manifestations of child undernutrition in India, the most prevalent co-occurrence being “only anemia”, followed by triple burden of co-occurrence of stunting, underweight and anemia (SUA). The reviewer believes that the results of this study demonstrate the importance of anemia as a CIAF and provide useful information in the assessment of malnutrition in children. However, there are several limitations in this study.
1. The reviewer feels that the results of this study are nothing more than a compilation of data and a list of numbers. The reviewer believes that scientific article should be analysed using appropriate statistical analysis and published based on objective findings.
2. The reviewer considers this article to have numerous problems with its presentation as a scientific article. The “Introduction” section of this paper is very long. The “Introduction” section of this article contains several sentences that deviate from the content of this study. The “Introduction” section should be limited to only what is necessary for this study.
3. Similarly, the “Results” and “Conclusion” sections are too long. The “Results” section should objectively present only the results obtained from the appropriate statistical analysis. The “Conclusion” section should concisely present the most important findings of this study.
4. On the other hand, this paper lacks explanation in the “Methods” section. In particular, the reviewer cannot understand what population was targeted in this study or what was analyzed. The “Methods” section should explain to detail that readers would be able to repeat the measurements after reading this article.
5. The authors conclude that the findings of this study contribute to the understanding burdens and clustering of undernourishment at various levels, which can help develop screening measures to identify such high-risk individuals or districts and and target interventions for marginalized groups accordingly. However, the reviewer considers that the study design and analytical methods used in this study do not lead to the development of a screening measures.
6. How do the authors plan to develop the findings from this study in the future? Based on the content of this article, the reviewer cannot understand the potential for further development of this study.
7. Why the figures and tables in this paper start with Figure 3 and Table 3, respectively? Figures and tables should be numbered in the order in which they are described.
8. The figures and tables in this article do not have each units or scales. The reviewer cannot understand what data is being presented because there are no units or scales.
Author Response
S.no. |
Comment |
Response |
|
This study aimed to attempt to modify it by inclusion of a fourth form of undernutrition, that is anemia, serving as a proxy for micronutrient deficiencies among under-five children of India. As the results, the modified index of “composite Index of anthropometric failure (CIAF) + Anemia” identified thirteen manifestations of child undernutrition in India, the most prevalent co-occurrence being “only anemia”, followed by triple burden of co-occurrence of stunting, underweight and anemia (SUA). The reviewer believes that the results of this study demonstrate the importance of anemia as a CIAF and provide useful information in the assessment of malnutrition in children. However, there are several limitations in this study. |
Dear reviewer, thank you so much for your thorough review. Your comments have given us an opportunity to enhance this research study. |
1 |
The reviewer feels that the results of this study are nothing more than a compilation of data and a list of numbers. The reviewer believes that scientific article should be analysed using appropriate statistical analysis and published based on objective findings. |
Thank you for your feedback. We definitely don’t want to give away such an impression. As a matter of fact, we have used statistical analysis to study variation in the co-occurrences of child undernutrition outcomes across different background characteristics using a bi-variate analysis that involves a chi-square test of association. But we do realise that we have missed mentioning it. We have also used local Moran’s I indicator for showcasing spatial autocorrelation and stated p-value for its statistical significance.
Now, based on your suggestions, we have modified the methods section and the findings section, where we clearly indicate the use of statistical methods and p-value and provide scientific insights. We have also added regression analysis results to strengthen our study (Table 5; Table 9 in the main text, and Table S2 in the supplementary section). Further, spatial analysis for the quadruple burden of child undernutrition has been incorporated too (Table 8 and Table 9). |
2 |
The reviewer considers this article to have numerous problems with its presentation as a scientific article. The “Introduction” section of this paper is very long. The “Introduction” section of this article contains several sentences that deviate from the content of this study. The “Introduction” section should be limited to only what is necessary for this study. |
Thank you for pointing this out. The unnecessary parts that deviated from the study’s objective have now been removed. It is now made more crisp and short pertaining to only what is necessary for the study. |
3 |
Similarly, the “Results” and “Conclusion” sections are too long. The “Results” section should objectively present only the results obtained from the appropriate statistical analysis. The “Conclusion” section should concisely present the most important findings of this study. |
Thank you for your feedback. We have shifted some of the descriptive statistics, which do not involve much statistical analysis, to the supplementary material section for shortening the results section. We also have shortened the conclusion section as suggested. |
4 |
On the other hand, this paper lacks explanation in the “Methods” section. In particular, the reviewer cannot understand what population was targeted in this study or what was analyzed. The “Methods” section should explain to detail that readers would be able to repeat the measurements after reading this article. |
Agree. The target population is the children under the age of five. As suggested, we have enhanced the methods section, making it more detailed so that it can be easy for others to understand. |
5 |
The authors conclude that the findings of this study contribute to the understanding burdens and clustering of undernourishment at various levels, which can help develop screening measures to identify such high-risk individuals or districts and and target interventions for marginalized groups accordingly. However, the reviewer considers that the study design and analytical methods used in this study do not lead to the development of a screening measures. |
Agree. The word screening is not appropriate. The better word would be a “monitoring” measure rather than a “screening” measure. The vision that the authors have is that if we can have an indicator that can be created from the existing government data and monitoring mechanisms that can help us track down the most vulnerable children who are suffering from multiple undernutrition morbidities or specialised nutrition morbidities, then accordingly targeted interventions could be planned for such children catering to their special needs. We have now replaced the word “screening” with “monitoring” and have explained this in more detail in the text (587 to 598; Line 651 to 653). |
6 |
How do the authors plan to develop the findings from this study in the future? Based on the content of this article, the reviewer cannot understand the potential for further development of this study. |
There is further scope of research in addition to this study. This section is also now enhanced in the writing (Line 622 to 632). |
7 |
Why the figures and tables in this paper start with Figure 3 and Table 3, respectively? Figures and tables should be numbered in the order in which they are described. |
Thank you so much for pointing this out. It is corrected now. |
8 |
The figures and tables in this article do not have each units or scales. The reviewer cannot understand what data is being presented because there are no units or scales. |
Thank you for sharing this. Now, we have ensured scales and units at each of the places where it was missing. |
Reviewer 3 Report
Comments and Suggestions for Authors
The article addresses a topic of great interest to the readership of Nutrients, namely the co-occurrence of different forms of childhood malnutrition in the population of India. An important conclusion is drawn, emphasizing the need for early identification and specialized treatment of affected children to prevent the onset of scarring effects.
The manuscript is well-written and includes all the essential sections of a scientific article. However, we believe that some modifications could enhance the study:
-
Introduction: While the introduction is well-crafted, it is relatively brief, which limits the reader's ability to fully understand the current state of research on this topic. Additionally, although the references cited are appropriate, they are somewhat outdated. The obsolescence index of the references (median publication age) is 7 years. We recommend updating the references with more recent studies.
-
Methodology: The methodology section is too concise and lacks sufficient detail to ensure reproducibility of the study. Expanding this section would improve the transparency and reliability of the work.
-
Results: The results are appropriately presented and supported by clear and illustrative tables and figures. However, we recommend reducing the font size of the numbers and text in Figures 3.4 and 3.5 to improve readability. Additionally, we suggest including explanations of the acronyms in the table footnotes (e.g., CIAF in Table 3.3; SUA, SA in Table 3.4).
-
Reference Issue: In line 129, the citation "Khalig et al., 2022" appears, but no such reference is included in the bibliography. This should either be removed or replaced with reference 23, followed by renumbering the subsequent citations.
-
Discussion: The discussion is too brief, limiting the depth of comparison between the study's findings and previous research. We recommend adding a paragraph outlining the study’s strengths, limitations, and its main contribution in comparison to prior studies.
-
Conclusions: The conclusions are overly lengthy, and some of their content might be better placed within the discussion section. Streamlining this section would improve the overall structure and readability of the article.
Author Response
S.no. |
Comment |
Response |
|
The article addresses a topic of great interest to the readership of Nutrients, namely the co-occurrence of different forms of childhood malnutrition in the population of India. An important conclusion is drawn, emphasizing the need for early identification and specialized treatment of affected children to prevent the onset of scarring effects. The manuscript is well-written and includes all the essential sections of a scientific article. However, we believe that some modifications could enhance the study: |
Thank you so much for sharing your precious feedback for further strengthening of our study. |
1 |
Introduction: While the introduction is well-crafted, it is relatively brief, which limits the reader's ability to fully understand the current state of research on this topic. Additionally, although the references cited are appropriate, they are somewhat outdated. The obsolescence index of the references (median publication age) is 7 years. We recommend updating the references with more recent studies. |
Thank you for the comments. As suggested, we have modified the introduction with parts that are more related to the current objective of the study. Additionally, we have now added references to more recent studies. |
2 |
Methodology: The methodology section is too concise and lacks sufficient detail to ensure reproducibility of the study. Expanding this section would improve the transparency and reliability of the work. |
Thank you so much for pointing this out. As suggested, we have now made the methods section more detailed. |
3 |
Results: The results are appropriately presented and supported by clear and illustrative tables and figures. However, we recommend reducing the font size of the numbers and text in Figures 3.4 and 3.5 to improve readability. Additionally, we suggest including explanations of the acronyms in the table footnotes (e.g., CIAF in Table 3.3; SUA, SA in Table 3.4). |
Thank you for your comment. As suggested, we have reduced the font size in the 2 figures (now Figure 4 and Figure 5) to improve readability. Also, in Figure 4, for ease of understanding, we have now kept only the eight most prevalent co-occurences instead of all the fourteen categories of the new index of CIAF+Anemia. To reduce complexity, we have also adhered to using only data from the three rounds of NFHS as a source to show the trend and removed the prevalences of the CNNS survey to avoid any confusion. Similarly, in Figure 4, we have only kept the NFHS as a data source and removed the prevalence of the CNNS survey to reduce complexity. I have added the bar of “India” as well which was earlier not there. In addition, explanations on the acronyms in a detailed manner have been now added in the methods section itself since these acronyms are used throughout the study. |
4 |
Reference Issue: In line 129, the citation "Khalig et al., 2022" appears, but no such reference is included in the bibliography. This should either be removed or replaced with reference 23, followed by renumbering the subsequent citations. |
Thank you so much for pointing out this. It is a paper by Asif Khaliq which has now been added to the references (Reference number 15). As suggested, renumbering of the references is done. |
5 |
Discussion: The discussion is too brief, limiting the depth of comparison between the study's findings and previous research. We recommend adding a paragraph outlining the study’s strengths, limitations, and its main contribution in comparison to prior studies. |
Thank you for your feedback. As suggested, we have now elongated the discussion section and added more on the study’s main contribution, strengths and limitations (Line 571 to line 632). |
6 |
Conclusions: The conclusions are overly lengthy, and some of their content might be better placed within the discussion section. Streamlining this section would improve the overall structure and readability of the article. |
Thank you so much for the suggestion. As guided, we have now streamlined the text from the conclusion and discussion section to improve the readability of the article. The conclusion is now shortened. |
Round 2
Reviewer 1 Report
Comments and Suggestions for Authors
in my opinion the work is well written, thank you for the updates
Comments on the Quality of English Languagethe work can be published
Author Response
I sincerely appreciate your time and thoughtful review. Thank you!
Reviewer 2 Report
Comments and Suggestions for Authors
I think all responses to reviewers' comments have been addressed satisfactorily.
I have no comments on the revised manuscript.
Author Response

(The authors gave the same response as above.)

Reviewer 3 Report
Comments and Suggestions for Authors
Is ok now
Author Response

(The authors gave the same response as above.)
